# Effects of Liquid Yucca Supplementation on Nitrogen Excretion, Intestinal Bacteria, Biochemical and Performance Parameters in Broilers

**DOI:** 10.3390/ani9121097

**Published:** 2019-12-09

**Authors:** Mousa M. Ayoub, Hamada A. Ahmed, Kadry M. Sadek, Mahmoud Alagawany, Mohamed E. Abd El-Hack, Sarah I. Othman, Ahmed A. Allam, Mervat A. Abdel-Latif

**Affiliations:** 1Department of Animal Hygiene and Zoonoses, Faculty of Veterinary Medicine Damanhour University, Damanhour 22511, Egypt; mousa.ayoub@vetmed.dmu.edu.eg; 2Department of Nutrition and Veterinary Clinical Nutrition, Faculty of Veterinary Medicine, Damanhour University, Damanhour 22511, Egypt; hamada_nutrition@vetmed.dmu.edu.eg; 3Department of Biochemistry, Faculty of Veterinary Medicine, Damanhour University, Damanhour 22511, Egypt; kadry.sadek@vetmed.dmu.edu.eg; 4Poultry Department, Faculty of Agriculture, Zagazig University, Zagazig 44511, Egypt; dr.mohamed.e.abdalhaq@gmail.com; 5Biology Department, College of Science, Princess Nourah bint Abdulrahman University, Riyadh 11671, BO. Box 24428, Saudi Arabia; sialothman@pnu.edu.sa; 6Department of Zoology, Faculty of Science, Beni-suef University, Beni-suef 65211, Egypt; allam1081981@yahoo.com

**Keywords:** ammonia, broiler, intestinal bacteria, *Yucca schidigera*

## Abstract

**Simple Summary:**

*Yucca schidigera* had a positive effect on the improvement of economic traits, performance, and carcass characteristics of broilers. Saponin is the main steroidal chemical constituent of *Yucca schidigera* extract, which physically binds ammonia and reduces its level. Use of natural antibiotic alternatives such *Yucca schidigera* is necessary to improve growth rates and feed utilization, as well as decreasing nitrogen losses, feed cost, and global environmental pollution.

**Abstract:**

This study was done to determine the impacts *Yucca schidigera* supplementation to drinking water on the excretion of nitrogen, and subsequently the level of ammonia, intestinal bacterial count, hematological and biochemical parameters, and some performance parameters. A total of 270 one-day old Cobb 500 chicks were equally divided into three groups (90 chicks/group). The first control group (G1) was fed on the basal diets without any yucca supplementation. The 2nd and 3rd groups (G2 and G3) were fed on basal diets with Yucca Plus liquid^®^, at an 8 h/day supplementation rate of 0.5, and 1 mL/L to drinking water, respectively. The chicks that received yucca showed significant decreases in litter nitrogen content, when compared to controls. The chicks that received liquid yucca had reduced counts of total bacteria (TBC) (*p* < 0.05), *Escherichia coli*, and a non-significant increase in the number of lactic acid producing bacteria. They also showed increased activity of antioxidant enzymes, increased levels of immunoglobulins M and G, and decreased levels of lipid peroxidation biomarkers, without a harmful effect on liver and kidney function. The chicks that received yucca showed a better feed conversion ratio. In conclusion, the use of natural additives is necessary to decrease nitrogen losses, feed cost, and environmental pollution; without adverse impacts on animal performance. Liquid supplementation of saponins is valuable for the performance, gut health, and welfare of broiler chickens.

## 1. Introduction

It is well-known that the cost of feeding represents nearly 70% of the total production cost of broilers. The intensive breeding and high protein diet causes many environmental and intestinal hazards. For example, high aerial ammonia increases the incidence of enteric diseases. Higher concentrations of gaseous NH_3_ is the most predominant pollutant in poultry farms, which adversely affects performance, the welfare of birds, and consequently, human health [1]. In addition, high levels of ammonia may reduce feed intake and deteriorate the growth rate of birds due to damage to the respiratory tract, increasing their susceptibility to Newcastle disease virus, the incidence of air sacculitis and kerato conjunctivitis, in addition to the already high prevalence of *Mycoplasma gallisepticum* [2]. Decreasing nitrogen emissions in poultry houses is important to keep both birds and humans healthy [3]. About 70% of nitrogenous substances in excrement originate from urine, and 30% from feces. Poultry excreta contains about 60–65% uric acid, 10% ammonium salts, 2–3% urea. Remaining creatinine, and especially uric acid, is rapidly changed to NH_3_ by microbes [4].

Several chemicals and treatments have been adopted to control ammonia emissions in poultry houses such as zeolites, aluminum chloride, and supplementation of *Yucca schidigera* extract in poultry feed [5,6,7]; these compounds also improve metabolic efficiency, egg weight, feed conversion, and production traits [8,9,10,11].

*Yucca schidigera* is a widespread herbal plant with different beneficial activities, such as growth stimulation and immunostimulation, as well as antioxidant, anti-inflammatory, anticarcinogenic effects, and hypoglycemic and hypocholesterolemic activities [12]. *Y. schidigera* plays a key role in mitigating ammonia emissions and fecal odors emanating from poultry houses and surrounding areas [13]. Furthermore, the extract and powder of *Y. schidigera* are abundant in steroidal saponins, and are used as feed supplements and cosmetic [14]. *Yucca schidigera* is a commercially important source of various enzymes, saponins, antioxidants, and resveratrol [7]. Saponin is the main steroidal chemical constituent of *Y. schidigera* extract, which physically binds ammonia and reduces its level. Its extract contains a glycol-fraction, which has ammonia binding capabilities, and a saponin fraction, which has antiprotozoal and antimicrobial properties. *Yucca schidigera* has a positive effect on the improvement of economic traits, performance, and carcass characteristics of broilers and quails [15,16]. The health status of the intestinal tract shows a significant improvement, and the growth of pathogenic bacteria was reduced as a result of *Yucca schidigera* administration [17]. Therefore, the aim of this study was to evaluate the effect of adding *Yucca schidigera* to the drinking water of broiler chickens on reducing the atmospheric ammonia levels, and minimizing litter nitrogen. Also, the study aimed to assess its effects on intestinal bacterial flora, biochemistry, and other performance parameters.

## 2. Materials and Methods

All procedures and experiments were performed in accordance with the Ethics of the Committee of Local Experimental Animal Care, and were approved by the Nutrition and Veterinary Clinical Nutrition Institutional Committee, Faculty of Veterinary Medicine, Damanhour University, Damanhour, Egypt (DMU2018-0045). All efforts were made to minimize animal suffering.

### 2.1. Birds, Housing and Vaccinations

A total of 270 one-day old (DO), commercial, unsexed Cobb 500 chicks, obtained from a local commercial hatchery were equally divided into three groups (90 chicks/group; each group had six replicates each of 15 chicks). Birds were raised on deep litter and received experimental diets for five consecutive weeks. The ambient temperature was maintained at 32 °C in the first week, and gradually decreased (3 °C/week) to 21 °C on the 5th week. Chicks were exposed to continuous light during the first two days of age, and then exposed to light for 23 h, followed by an hour of darkness per day thereafter. Diets and fresh water were supplied *ad libitum*.

### 2.2. Diets and Experimental Design

The chicks were fed on starter diet during the first three weeks (0 to 21 days) and finisher diet during the second two weeks (22 to 35 days), to meet the recommendation of the National Research Council [18] for broilers. The diets were chemically analyzed as shown in Table 1. The first control group (G1) was fed on the basal diet without any yucca supplementation in water, while the 2nd and 3rd groups (G2 and G3) were fed on basal diets with yucca supplementation (Yucca Plus liquid^®^, Beijing Xiqin Pharmaceutical Co., Ltd., Beijing, China), at an 8 h/day supplementation rate of 0.5, and 1 mL/L to drinking water, respectively.

### 2.3. Performance Parameters

The initial body weight (iBW), final body weight (fBW), feed intake (FI), and body weight gain (BWG) were recorded. Whereas, the feed conversion ratio (FCR) and protein efficiency ratio (PER) were calculated.

### 2.4. Litter Sampling and Analysis

The litter was sampled on the 21st and 35th days from each poultry pen. Samples from 10 random locations (avoiding areas around feeders and drinkers) were collected. Subsequently, the random litter samples were thoroughly mixed in a plastic bag, and 250 g was weighed and delivered for further processing in the laboratory. The sample was ground to pass through a 2 mm sieve, and frozen until further analysis. Dry matter (DM) of the litter was determined by oven-drying at 105 °C for 48 h, and calculating the differences in weight. Ash contents of litter samples were determined by incineration at 550 °C overnight. Nitrogen in the litter samples was determined by using the Kjeldahl method, according to Association of Official Analytical Chemists (AOAC) [19] (method 988.05).

### 2.5. Evaluation of Intestinal Bacterial Flora

At 10, 18 and 28 days of age, thirty six birds were randomly selected and slaughtered. In order to evaluate the effect of yucca supplementation at different concentrations to drinking water on the colonization of pathogenic and beneficial bacteria in comparison to the control group, a total of 36 cecal samples were collected from all chickens, throughout the experimental period (12 samples/group). Cecum content weighing 1 g was removed from the birds. To determine the colony forming units (CFU), one gram of the cecum content in the vicinity of a hot flame was added to 9 mL of peptone water (tube number one), and the solution was shaken. Then, 1 mL of solution was added to the next tube (tube number two), by a sampler containing 9 mL of sterile peptone water. This operation was conducted serially until tube number eight, and a dilution series was prepared according to Mountzouris et al. [20]. The samples were placed on a plate containing eosin methylene blue agar, MacConkey agar, and Rogasa medium for the growth of *Escherichia coli*, coliforms, and lactic acid bacteria. However, nutrient agar medium was prepared for the counting of the total cultured bacteria. This process was repeated for each sample. MacConkey, eosin methylene blue, and nutrient agar, as well as Rogasa media, were incubated at 37 °C for 24 h. Finally, the samples were evaluated when they reached between 25 and 300 colonies, which were selected at an appropriate dilution. After counting, the number of colonies was multiplied by the inverse of the dilution to obtain the number of bacteria.

### 2.6. Biochemical Parameters

At the third and the fifth weeks of age, six birds from each group were randomly selected and fasted overnight, blood samples were then collected through slaughtering into non-heparinized chilled tubes, followed by centrifugation at 3500 rpm for 15 min at 4 °C. The separated sera was kept at −20 °C for biochemical investigation. Lipid peroxidation (LPO), assessed as the generation of thiobarbituric acid-reactive substances (TBARS) [21]; glutathione (GSH), assessed by Ellman’s reagent [22]; catalase (CAT) (EC 1.11.1.6) activity, determined as the rate of hydrolysis of H_2_O_2_. The superoxide dismutase (SOD) activity was determined as the sample was reacted with an adrenaline solution, and the degree of prevention of adenochrome synthesis from auto-oxidation was assessed at 480 nm [23]. The activities of the serum alanine amino transferase (ALT), and levels of serum proteins, globulin, albumin, and creatinine were spectrophotometrically determined using industrially available kits (Biodiagnostic Co., Dokki-Giza, Egypt), following the manufacturer’s instructions. Serum immunoglobulins (IgG and IgM) were determined using commercial ELISA kits (Kamiya Biomedical Company, Tukwila, WA, USA), according to Bianchi et al. [24]. Triglycerides and total cholesterol were determined according to Fossati Prencipe [25], and Stein [26], respectively.

### 2.7. Statistical Analysis

The obtained data were subjected to an Analysis of Variance (ANOVA) test, appropriate for a completely randomized design, using the statistical program SPSS.20^®^ (IBM Cooperation, Armonk, NY, USA) to assess the significant differences with a Tukey’s range test. Statements of statistical significance were based on *p* < 0.05.

## 3. Results

### 3.1. Performance Measurements

Results concerning the effect of yucca supplementation at two levels (0.5 and 1 mL/L of drinking water) on broiler performance are shown in Table 2. There was a numerical increase in final body weight in the groups treated with yucca, when compared to the control group. Concerning total body weight gain, the experimental groups supplemented with 0.5 and 1 mL/L showed a 1.65 and 2.32% increase, relative to control increase in weight gain. Table 2 shows the significant difference in total feed intake in groups with yucca supplementation. There was significant improvement in feed conversion ratio, and protein efficiency in the yucca supplemented group at level 0.5 mL/L drinking water, compared to the control one.

### 3.2. Nitrogen, Moisture and Ash Content of the Litter

Effect of yucca supplementation on different groups on litter nitrogen %, is shown in Table 3. At the 21st day, groups which were supplemented with yucca showed significant decreases in nitrogen content compared to untreated one (0.72 ± 0.023 _b_ and 0.52 ± 0.023 _c_, vs. 0.813 ± 0.013 _a_) for 0.5 and 1 mL/L vs. control. In the same manner, litter samples of groups supplemented with yucca at levels of 0.5 and 1 mL/L at the 35th day showed significant decreases in nitrogen content comparing to the control. Groups supplemented with yucca at levels of 0.5 and 1 mL/L at the 35th day showed a lower nitrogen content by about 60 and 46.67%, compared to the control. Concerning the effect of yucca supplementation on litter moisture content, during the 3rd and the 5th weeks, yucca supplemented groups showed significant decreases in moisture content, compared to the control group.

### 3.3. Evaluation of Intestinal Bacterial Count

The results in Table 4 show that ceacal samples taken at 10 days of age had a significant decrease in total colony counts (*p* < 0.05) in the group that received yucca at rate of 1 mL/L, as compared to the control group, and a non-significant decrease in the group receiving 0.5 mL yucca/L. Also, a significant decrease in the count of *E. coli* (*p* < 0.05) in both yucca groups was detected, compared to the control group. In addition, no significant change in the count of lactic acid producing bacteria (*p* > 0.05) in either yucca group was recorded. At 18 days of age, a numerical decrease in the total colony count (*p* > 0.05) in the two yucca groups was found. Supplementing yucca to the drinking water also led to a significant decrease in *E. coli* numbers (*p* < 0.05). However, the number of lactic acid producing bacteria was not affected.

At 28 days of age, the total colony count was insignificantly decreased in the group receiving 0.5 mL yucca/L, as compared to the control group. Furthermore, supplementing yucca insignificantly depressed the number of *E. coli* (*p* > 0.05) and lactic acid producing bacteria (*p* > 0.05) in both yucca groups, in comparison to the control group.

### 3.4. Biochemical Parameters

The results revealed that the addition of yucca significantly (*p* < 0.05) increased the activity of antioxidant enzymes (SOD and GSH), and decreased the level of malondialdehyde (MDA) (a lipid peroxidation biomarker), compared to the control group at the end of the study. In addition, serum globulin and IgM were insignificantly higher in yucca supplemented groups than that of the control group on the 21st and 35th days of age. However, IgG was lower in G3 compared to G1 and G2 (*p* = 0.001). On the other hand, kidney and liver function biomarkers were not affected.

## 4. Discussions

Our results concerning growth performance, agree with that of Cabuk et al. [27], who found a numerical increase in the body weight of broilers fed a diet supplemented with *Yucca schidigera,* compared to a control group. This may be related to the positive effects of steroid saponin present in yucca on nutrient absorption. Previous research has demonstrated that saponin can improve the absorption of nutrients from the intestinal tract [28]. These results are supported by those of Su et al. [29], who found that dietary supplementation of *Y. schidigera* powder at a level of 100 ppm improved body weight gain and feed efficiency during the finisher period. In line with these results, Sahoo et al. [30] found that yucca supplementation could effectively lead to a better feed conversion rate and protein efficiency ratio, than in an un-supplemented group. The improvement of FCR and PER in the yucca supplemented groups may be attributed to the presence of natural saponin from *Y. schidigera*, which might result in the emulsification of oil fats, promoting their digestion, and the absorption of vitamins and minerals [31].

In poultry, about 60–70% of the excreted nitrogen is in the form of uric acid, which starts to be converted into urea as soon as urine comes into contact with feces, by the action of urease. The conversion rate of urea into ammonia is temperature dependent, and is greatly decreased at temperatures below 5–10 °C [32]. This result confirms that Yucca Plus liquid^®^ added to drinking water can enhance the absorption of nitrogen and reduce its excretion, and therefore reduce the level of ammonia in the animal’s digestive tract and excreta [33]. This reduces the level of ammonia in the poultry houses, which is detrimental to the performance of modern commercial broilers [34]. Moreover, low percentages of nitrogen and moisture have been found in the yucca treated groups, compared to the control one [30]. This may be due to the amount of saponin in yucca, which has surfactant properties, is enough to increase nutrient absorption [10,35,36,37,38]. Also, yucca is used as an NH_3_ inhibitor, which can improve the air quality around the birds and improves their performance [5]. Also, our results agree with those of Cheeke [39], who concluded that *Y. schidigera* extracts were capable of binding NH_3_ directly, thus decreasing its production in animal houses. Similarly, the addition of yucca at 100 ppm has been shown, in comparison with the other levels (0, 50, 100, and 200 ppm), to significantly reduce emissions (*p* < 0.05) of NH_3_ by 44 and 28% on the 1st and 2nd days of manure storage, respectively [40]. Also, Hussain et al. [41] reported that extracts of yucca have ammonia-binding properties. It was hypothesized that by binding ammonia in the caecum, yucca could affect utilization of crude protein and dietary urea in ceca. The impact of yucca extract supplementation is manifested in mitigating levels of ammonia in the caecum of animals [42].

The results in Table 4 indicated that the addition of yucca to the broiler drinking water is of value in reducing total bacterial count and the number of *E. coli* in different ages, especially in the young age. Wang and Kim [28] found that *E. coli* counts were linearly inhibited by *Yucca schidigera* extract treatments, compared with the non-treated group at both five and eight weeks, and no difference was observed on the *Lactobacillus* population throughout the experimental period.

The result of ALT and creatinine revealed that yucca had no adverse effect on liver and kidney functions. Also, the levels of IgM and G revealed that yucca had an immunostimulant effect in broilers. Similar findings were obtained by Su et al. [29], who found that dietary supplementation of 100 ppm *Y. schidigera* powder increased IgG, and induced IgM. At a level of 200 ppm, a better effect on cellular and humoral immune responses in broilers was seen. Our results showed an adjustment effect of yucca extract on the humoral and cellular immune responses.

Regarding antioxidative biomarkers, results in Table 5 confirmed that the addition of yucca improved the activity of antioxidant enzymes including SOD, CAT, and GSH, and decreased lipid peroxidation biomarkers compared to the control group. Similarly, Su et al. [29] demonstrated that broiler chickens fed yucca showed a significant improvement in SOD activity, and exhibited a strong antioxidative effect. Yucca is a good source of many phytochemicals like resveratrol and other bioactive components, such as yuccaols (A, B, C, D, and E) [10,11]. Resveratrol derived from yucca possess potent anti-inflammatory and antioxidant effects [9,40]. The extract or powder of yucca leaves or roots could exhibit strong antioxidant properties [5]. The same results were reported by Alagawany et al. [10], who found that the activity of SOD and GSH were quadratically improved in yucca (with 50, 100, or 150 mg/kg of yucca extract) supplemented groups. The concentration of MDA was decreased with yucca supplementation in comparison with a control group.

## 5. Conclusions

Yucca appeared to decrease nitrogen excretion, thus improving litter quality and bird welfare, and consequently, improving the gut health and performance of broiler chickens. This study gives new evidence for the use of saponins as natural alternatives to antibiotics and growth promoters.

## Figures and Tables

**Table 1 animals-09-01097-t001:** Ingredient composition of the experimental basal diets.

Ingredients	Starter (0–21)	Finisher (22–35)
Yellow Corn	55.60	68.36
Soybean meal, 46%	36.1	21.5
Full-fat soya bean	4.5	7
DL-Methionine	2.4	1.5
L-Lysine	1.6	1.4
Mono-calcium phosphate	1	0.75
Calcium carbonate	1.7	1.4
Sodium chloride	0.27	0.27
Vitamin and mineral premix *	0.3	0.3
Choline chloride	0.05	0.05
Physical anti-mycotoxin	0.05	0.05
Anticoccidials	0.05	0.05
Biological anti-mycotoxin	0.025	0.025
Analyzed and calculated composition (%)		
Metabolizable energy (ME) kcal/kg diet	3010	3175
Crude protein (CP)	22.4	18
Ether extract (EE)	3	4.3
Crude fiber (CF)	2.18	2.8
Calcium (Ca)	1.04	0.85
Available Phosphorous	0.51	0.46
Linoleic acid	1.5	2
Methionine	0.58	0.45
Methionine + Cysteine	0.98	0.79
Lysine	1.44	1.1
Sodium (Na)	0.2	0.2
Chloride (Cl)	0.27	0.26

* Each 2 kg of vitamin mineral premix contains: vitamin A, 1,200,000 IU; vitamin D_3_, 300,000 IU; vitamin E, 700 mg; vitamin K_3_, 500 mg; vitamin B_1_, 500 mg; vitamin B_2_, 200 mg; vitamin B_6_, 600 mg; vitamin B_12_, 3 mg; folic acid, 300 mg; choline chloride, 1000 mg; Niacin, 3000 mg; Methionine, 3000 mg; Biotin, 6 mg; panathonic acid, 670 mg; manganese sulphate, 3000 mg; iron sulphate, 10,000 mg; zinc sulphate, 1800 mg; copper sulphate, 3000 mg; iodine, 1.868 mg; cobalt sulphate, 300 mg; and selenium, 0.108 mg.

**Table 2 animals-09-01097-t002:** Effect of dietary yucca supplementation on performance parameters of broiler chickens.

Parameter	Group ^1^			*p*-Value ^2^
	G1	G2	G3	
Body weight (iBW, g)	174.46 ± 2.26	173.04 ± 2.31	173.15 ± 2.23	0.887
Final body weight (fBW, g)	1832.28 ± 26.21	1857.07 ± 34.13	1869.46 ± 28.56	0.669
Body weight gain (BWG, g)	1657.83 ± 23.99	1684.02 ± 31.87	1696.3 ± 26.37	0.604
Total feed intake (TFI, g)	3172 ^a^ ± 15.72	3166.5 ^b^ ± 23.98	2809.3 ^c^ ± 18.95	0.005
Feed conversion ratio (FCR)	1.93 ± 0.02 ^a^	1.91 ± 0.04 ^a^	1.67 ± 0.03 ^b^	0.05
Protein efficiency ratio (PER)	2.41 ± 0.05 ^b^	2.44 ± 0.05 ^b^	2.88 ± 0.05 ^a^	0.05

^a–c^ Means within the same row that carry different superscripts are significantly different at *p* < 0.05. ^1^ G1: control without supplementation, G2: 0.5 mL/L, 8 h/day drinking water and G3: 1 mL/L, 8 h/day drinking water, Mean ± SE. ^2^ Overall treatment *p*-value.

**Table 3 animals-09-01097-t003:** Effect of liquid yucca supplementation on litter content of nitrogen, moisture, and ash.

Parameter	Group ^1^			*p*-Value ^2^
	G1	G2	G3	
Nitrogen (%)				
Day 21	0.813 ± 0.013 ^a^	0.72 ± 0.023 ^b^	0.52 ± 0.023 ^c^	0.005
Day 35	1.20 ± 0.046 ^a^	0.72 ± 0.046 ^b^	0.56 ± 0.023 ^c^	0.005
Moisture (%)				
Day 21	33.50 ± 0.12 ^a^	31.97 ± 0.15 ^b^	30.70 ± 0.06 ^c^	0.005
Day 35	32.97 ± 0.15 ^a^	30.77 ± 0.15 ^b^	29.97 ± 0.20 ^c^	0.005
Ash (%)				
Day 21	22.80 ± 0.12 ^a^	22.23 ± 0.15 ^b^	22.17 ± 0.17 ^b^	0.039
Day 35	21.33 ± 0.20 ^a^	20.43 ± 0.22 ^b^	20.30 ± 0.15 ^b^	0.018

^a–c^ Means within the same row that carry different superscripts are significantly different at *p* < 0.05. ^1^ G1: control without supplementation, G2: 0.5 mL/L, 8 h/day drinking water, and G3: 1 mL/L, 8 h/day drinking water, Mean ± SE ^2^ Overall treatment *p*-value.

**Table 4 animals-09-01097-t004:** Effect of dietary yucca supplementation on the count of intestinal bacteria (total colony count (TCC), *E. coli* and Lactic acid producing bacteria (L.A.B.).

Parameter	Group ^1^			*p*-Value ^2^
	G1	G2	G3	
Day 10				
TCC × 10^9^	7.4 ± 0.32 ^a^	6.5 ± 0.39 ^a,b^	6.3 ± 0.40 ^b^	0.018
*E. coli* × 10^7^	8.2 ± 0.43 ^a^	6.3 ± 0.36 ^b^	6.1 ± 0.37 ^b^	0.017
L.A.B. × 10^8^	1.4 ± 0.23	1.9 ± 0.19	1.8 ± 0.23	0.751
Day 18				
TCC × 10^9^	6.4 ± 0.44	6.1 ± 0.33	6.0 ± 0.35	0.597
*E. coli* × 10^7^	2.5 ± 0.28 ^a^	1.0 ± 0.08 ^b^	1.2 ± 0.13 ^b^	0.016
L.A.B. × 10^8^	4.2 ± 0.47	5.1 ± 0.38	4.5 ± 0.35	0.743
Day 28				
TCC × 10^9^	6.0 ± 0.37 ^a^	4.0 ± 0.33 ^b^	6.0 ± 0.28 ^a^	0.015
*E. coli* × 10^7^	4.4 ± 0.39	3.5 ± 0.30	4.0 ± 0.32	0.653
L.A.B. × 10^8^	4.0 ± 0.32	4.5 ± 0.34	4.2 ± 0.47	0.721

^a,b^ Means within the same row that carry different superscripts are significantly different at *p* < 0.05. ^1^ G1: control without supplementation, G2: 0.5 mL/L, 8 h/day drinking water, and G3: 1 mL/L, 8 h/day drinking water, Mean ± SE. ^2^ Overall treatment *p*-value. TCC, total colony count; LAB, lactic acid bacteria.

**Table 5 animals-09-01097-t005:** Effect of dietary yucca supplementation on some blood parameters of broiler chickens.

Item	Group ^1^			*p*-Value ^2^
	G1	G2	G3	
At 21st day				
IgM (mg/100 mL)	74.87 ± 0.22	77.68 ± 0.87	82.51 ± 0.38	0.810
IgG (mg/100 mL)	183.86 ± 2.37	183.01 ± 1.89	184.46 ± 5.24	0.992
SOD (mg/dL)	37.84 ± 2.38 ^b^	40.16 ± 1.5 ^b^	56.35 ± 1.88 ^a^	0.001
CAT (mg/dL)	24.88 ± 1.74 ^b^	26.91 ± 1.8 ^b^	32.20 ± 0.84 ^a^	0.016
MDA (mg/dL)	62.83 ± 2.61	57.97 ± 0.12	57.70 ± 1.17	0.567
GSH (mg/dL)	37.02 ± 1.55	41.2 ± 1.47	40.61 ± 0.69	0.613
ALT (U/L)	36.07 ± 1.49	34.84 ± 2.11	38.69 ± 1.21	0.941
Creatinine (mg/dL)	0.47 ± 0.04	0.46 ± 0.04	0.47 ± 0.01	0.971
Albumin (g/dL)	1.99 ± 0.07	2.06 ± 0.08	2.08 ± 0.06	0.732
Total protein (g/dL)	5.3 ± 0.07	5.36 ± 0.11	5.21 ± 0.09	0.578
Globulin (g/dL)	3.33 ± 0.12	3.36 ± 0.19	3.13 ± 0.14	0.623
TGS (mg/dL)	169.6 ± 16.21	143.11 ± 10.17	180.62 ± 17.41	0.376
Cholesterol (mg/dL)	247.06 ± 9.29	205.43 ± 12.99	254.12 ± 8.58	0.897
At 35th day				
IgM (mg/100 mL)	83.72 ± 2.26	80.01 ± 1.34	80.73 ± 1.57	0.899
IgG (mg/100 mL)	201.36 ± 4.63 ^a^	201.84 ± 4.79 ^a^	80.73 ± 1.57 ^b^	0.001
SOD (mg/dL)	48.02 ± 2.84	50.19 ± 1.18	50.52 ± 0.68	0.373
CAT (mg/dL)	38.34 ± 2.91 ^c^	52.52 ± 0.98 ^b^	63.65 ± 2.7 ^a^	0.001
MDA (mg/dL)	81.54 ± 1.79 ^a^	68.04 ± 2 ^b^	60.66 ± 0.42 ^c^	0.001
GSH (mg/dL)	53.05 ± 2.65 ^c^	72.14 ± 1.67 ^b^	81.66 ± 1.74 ^a^	0.001
ALT (U/L)	45.78 ± 2.04	45.41 ± 2.06	42.56 ± 2.69	0.750
Creatinine (mg/dL)	0.87 ± 0.03	0.84 ± 0.03	0.83 ± 0.05	0.836
Albumin (g/dL)	2.13 ± 0.06	2.06 ± 0.12	2.13 ± 0.08	0.721
Total protein (g/dL)	5.43 ± 0.06	5.56 ± 0.11	5.63 ± 0.09	0.597
Globulin (g/dL)	3.24 ± 0.06	3.5 ± 0.16	3.5 ± 0.12	0.653
TGS (mg/dL)	168.89 ± 16.53	182.76 ± 9.05	210.96 ± 15.62	0.456
Cholesterol (mg/dL)	223.17 ± 3.89 ^b^	254.53 ± 7.49 ^a^	224.44 ± 12.73 ^b^	0.018

^a–c^ Means within the same row that carry different superscripts are significantly different at *p* < 0.05. ^1^ G1: control without supplementation, G2: 0.5 mL/L, 8 h/day drinking water and G3: 1 mL/L, 8 h/day drinking water, Mean ± SE. ^2^ Overall treatment *p*-value. IgM, immunoglobulin M; IgG, immunoglobulin G; SOD, superoxide dismutase; CAT, catalase; MDA, malondialdehyde; GSH, glutathione; ALT, alanine amino transferase; TGS, triglycerides.

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
