# Peer review of "Effects of Liquid Yucca Supplementation on Nitrogen Excretion, Intestinal Bacteria, Biochemical and Performance Parameters in Broilers"

_animals, 2019, doi:10.3390/ani9121097_

Round 1
Reviewer 1 Report
The study to test Yucca schidigera to protect from ammonia is not new. Actually, it is not clear when chicks are placed in same farm, if Yucca can protect only one group or all chickens. Also, some very minor points should be checked like small and capital letters for body weight in Table 1. The vaccination program presented is quite heavy. Is this necessary?
correct lines in discussion 2010 to 2017 as both start with similar phrase. Also, authors should compare results of water to feed Yucca intake in order to provide their suggestion on most optimum level of supplementation.
Author Response
Reviewer 1#
The study to test Yucca schidigera to protect from ammonia is not new. Actually, it is not clear when chicks are placed in same farm, if Yucca can protect only one group or all chickens.
Answer: of course, the study is not new for the use of yucca to protect against ammonia but the study provides more explanation about the effect of yucca on gut microbiota, antioxidant and immune status of birds with different levels of yucca. Yes, birds were raised in the same place to and the effect of yucca supplementation to protect against ammonia is clarified by litter analysis in different experimental groups. Also, all groups are separated from each other with different replicates. Actually the direct contact between birds and litter can affect. Moreover, the habit of birds to eat their excreta can affect to clarify the effect of yucca in one group compared to other groups. This is considered a small design to be applied on a large scale of poultry production using deep litter system to protect against respiratory manifestations.
Also, some very minor points should be checked like small and capital letters for body weight in Table 1. The vaccination program presented is quite heavy. Is this necessary?
Deleted.
Correct lines in discussion 2010 to 2017 as both start with similar phrase. Also, authors should compare results of water to feed Yucca intake in order to provide their suggestion on most optimum level of supplementation.
Done… several improvements have been made in the revised version….Hope this version is proper for publication.
Reviewer 2 Report
I am sorry for the too late reply.
I read the manuscript entitled “Effects of liquid yucca supplementation on nitrogen excretion, intestinal bacteria, biochemical and performance parameters in broilers” for possible publication in Animals. The experimental design, methods, results, and discussion are appropriate. After rewrite the following points, this manuscript is suitable for publishing on this journal:
Overall: This manuscript must be checked by professional English editing service. Materials and Methods: The meaning of abbreviations, “G1, G2 and G3”, should be explained in the section. Also, the authors should describe more detailed explanation about how to give water. Materials and Methods: The description about numbers of selected birds to get cecal samples or blood samples, is confusing. For example, “36 birds in each age of 10, 18 and 28 days” or “36 birds totally in 3 ages”? Results: In the text of “3.1. Performance measurements”, descriptions about data in G2 and G3 may be opposite. The authors should check and revise it. L34: “of litter” should be inserted between “nitrogen content” and “when”. L57: “to NH3” should be moved between “changed” and “by”. L65: “and quails” should be inserted between “of broilers” and “[12, 13]”. The study of [13] were carried out in quails. L79-80: The authors should change from “C” to “°C”. L90: “Table 1,” should be changed to “Table 1.” L103-104: “BWG” and “FCR” should be spelled out. L116: “days” between “10” and “18” should be deleted. L144-145: “according to Bianchi et al. [21], triglycerides [22] and total cholesterol [23].” is not good. The authors should revise it. L165, 168: “3rd week” and “5th week” should be changed to “21st day” and “35th day”, respectively. L169: “at 35th day” should be inserted between “liter” and “showed”. L181: “increase” should be changed to “change”. L186: “at” should be changed to “At”. L197-204: Descriptions about IgM and IgG should be added. L225: “[29]” should be changed to “[Witter, E. (Swedich University of Agriculture Sciences), personal communications]”. And then, Personal communications and un-published data must not be included in the References section. L227: “[30] thereby” should be changed to “[30]. Thereby”. L232, 235: “NH” should be changed to “NH3”. L233: “Also, These” should be changed to “Also, these”. L238: “also lead to increase in the number of lactic acid producing bacteria” should be deleted. In this study, it is unproven. L247: “YE” should be spelled out. L247-248: “Also, the levels of IgM and IgG revealed that Yucca has immunostimulant effect in broilers.” should be deleted. The sentence is repeated one in L243-244. Table 1: “SOBM” should be spelled out. Also, “Vitamin and mineral Premix” and “Physical Antimycotoxin” should be changed to “Vitamin and mineral premix” and “Physical antimycotoxin”, respectively. Table 5: In IgM at 3rd week, p-value is 0.810. Therefore, difference between means in G1, G2 and G3 is not significant.
Author Response
Reviewer 2#
I am sorry for the too late reply.
I read the manuscript entitled “Effects of liquid yucca supplementation on nitrogen excretion, intestinal bacteria, biochemical and performance parameters in broilers” for possible publication in Animals. The experimental design, methods, results, and discussion are appropriate. After rewrite the following points, this manuscript is suitable for publishing on this journal:
Thank you very much for giving us a chance to improve our paper…thanks a lot for your great efforts and help.
Overall: This manuscript must be checked by professional English editing service.
Done as required
Materials and Methods: The meaning of abbreviations, “G1, G2 and G3”, should be explained in the section.
Done
Also, the authors should describe more detailed explanation about how to give water.
Materials and Methods: The description about numbers of selected birds to get cecal samples or blood samples, is confusing. For example, “36 birds in each age of 10, 18 and 28 days” or “36 birds totally in 3 ages”?
Corrected
Results: In the text of “3.1. Performance measurements”, descriptions about data in G2 and G3 may be opposite. The authors should check and revise it.
Corrected
L34: “of litter” should be inserted between “nitrogen content” and “when”
Adjusted
L57: “to NH3” should be moved between “changed” and “by”. L65: “and quails” should be inserted between “of broilers” and “[12, 13]”.
Adjusted
The study of [13] were carried out in quails. L79-80: The authors should change from “C” to “°C”.
Done
L90: “Table 1,” should be changed to “Table 1.” L103-104: “BWG” and “FCR” should be spelled out. L116: “days” between “10” and “18” should be deleted.
Done
L144-145: “according to Bianchi et al. [21], triglycerides [22] and total cholesterol [23].” is not good. The authors should revise it.
Corrected
L165, 168: “3rd week” and “5th week” should be changed to “21st day” and “35th day”, respectively.
Done
L169: “at 35th day” should be inserted between “liter” and “showed”. L181: “increase” should be changed to “change”.
Done
L186: “at” should be changed to “At”. L197-204: Descriptions about IgM and IgG should be added.
Added
L225: “[29]” should be changed to “[Witter, E. (Swedich University of Agriculture Sciences), personal communications]”. And then, Personal communications and un-published data must not be included in the References section.
Adjusted
L227: “[30] thereby” should be changed to “[30].
Changed
Thereby”. L232, 235: “NH” should be changed to “NH3”. L233: “Also, These” should be changed to “Also, these”.
Changed
L238: “also lead to increase in the number of lactic acid producing bacteria” should be deleted. In this study, it is unproven.
Done
L247: “YE” should be spelled out.
Done
L247-248: “Also, the levels of IgM and IgG revealed that Yucca has immunostimulant effect in broilers.” should be deleted. The sentence is repeated one in L243-244.
Deleted
Table 1: “SOBM” should be spelled out.
Done
Also, “Vitamin and mineral Premix” and “Physical Antimycotoxin” should be changed to “Vitamin and mineral premix” and “Physical antimycotoxin”, respectively.
Done
Table 5: In IgM at 3rd week, p-value is 0.810. Therefore, difference between means in G1, G2 and G3 is not significant.
You are right- corrected -Thanks a lot for your valuable comments.
Reviewer 3 Report
In this manuscript, the authors have presented the results of their studies on the effect of Yucca liquid plus supplementation on poultry performance, biochemical paremeters, gut microbial flora and nitrogen excretion.
The presented data are sound. However, there are a few points that need to be addressed
The hypothesis and aim of the present work was not clearly stated in the abstract or introduction. The knowledge gap of the present work is not clearly mentioned. Relevant statistical analysis should be mentioned as footnote of tables and figures. Authors are suggested to change the format of Tables to the journal format. Authors should elaborate the abbreviations at its first appearance in the manuscript. Example: BWG. Authors are suggested to include recent references in the manuscript instead of old references. Following references are my suggestions to include instead of 18, 19, and 20 references.
Lavanya et al., 2012. Acetone Extract from Rhodomyrtus tomentosa: A Potent Natural Antioxidant. Evidence-Based Complementary and Alternative Medicine. 8, 2012.
Manubolu et al., 2014. Protective effect of Actiniopteris radiata (Sw.) Link. against CCl4 induced oxidative stress in albino rats. Journal of Ethnopharmacology. 153 (3): 744-752.
Lavanya et al., 2019. Protective Effects of Ammannia baccifera Against CCl4-Induced Oxidative Stress in Rats. International journal of environmental research and public health. 16 (8). 1440.
Do the yucca liquid plus (commercially available?!) contain glycofraction and saponins? Do the authors analyzed or determined the composition of yucca liquid plus by analytical methods to confirm the yucca liquid composition? Authors are suggested to include few more evidences /references implying the non-toxicity nature of saponins and glycofractions in the introduction section. Statistical representation as footnote under tables is not clear. Should present it clearly. Authors are suggested to include in materials and methods section how they have determined the dosage of yucca liquid for supplementation to animals. How do authors provide a statement that yucca liquid supplementation increases the nitrogen intestinal absorption in animals without proper experimental evidence. Authors should justify this. Does the excess nitrogen retention in poultry products effect human when consumed? Authors are suggested to include organoleptic tests to determine the quality of chicken or poultry products supplemented with yucca liquid for human consumption. Authors should discuss the role of antioxidant parameters and lipid peroxidation and compare their results with others work in discussion part. Other citations are needed to strengthen the importance and relevance of the present results in the discussion section. The format of references is not uniform. Need to check for typographical errors, plagiarism, punctuation, and grammar throughout the manuscript.
Author Response
Reviewer 3#
In this manuscript, the authors have presented the results of their studies on the effect of Yucca liquid plus supplementation on poultry performance, biochemical paremeters, gut microbial flora and nitrogen excretion.
The presented data are sound. However, there are a few points that need to be addressed
The hypothesis and aim of the present work was not clearly stated in the abstract or introduction. The knowledge gap of the present work is not clearly mentioned.
Relevant statistical analysis should be mentioned as footnote of tables and figures.
Done
Authors are suggested to change the format of Tables to the journal format. Authors should elaborate the abbreviations at its first appearance in the manuscript. Example: BWG.
Corrected
Authors are suggested to include recent references in the manuscript instead of old references. Following references are my suggestions to include instead of 18, 19, and 20 references.
Ok all references have been added
Lavanya et al., 2012. Acetone Extract from Rhodomyrtus tomentosa: A Potent Natural Antioxidant. Evidence-Based Complementary and Alternative Medicine. 8, 2012.
Added
Manubolu et al., 2014. Protective effect of Actiniopteris radiata (Sw.) Link. against CCl4 induced oxidative stress in albino rats. Journal of Ethnopharmacology. 153 (3): 744-752.
Added
Lavanya et al., 2019. Protective Effects of Ammannia baccifera Against CCl4-Induced Oxidative Stress in Rats. International journal of environmental research and public health. 16 (8). 1440.
Added
Do the yucca liquid plus (commercially available?!) contain glycofraction and saponins?
Yes, Yucca plus liquid is commercially available as mentioned in the section of material and methods with the following composition:
Each 1 ml contains:
Yucca schidigera 0.1 ml
Vitamin A 1250 IU
Vitamin E 2.5 mg
Vitamin C 10 mg
Vitamin K3 0.6 mg
Propylene glycol 50 mg
Purified water up to 1 ml
Do the authors analyzed or determined the composition of yucca liquid plus by analytical methods to confirm the yucca liquid composition?
Unfortunately, we did not analyze the yucca liquid plus, but we depend on the previous studies related to our study.
Authors are suggested to include few more evidences /references implying the non-toxicity nature of saponins and glycofractions in the introduction section.
We added some useful information about yucca and saponin. Thank you very much for your observation.
Statistical representation as footnote under tables is not clear. Should present it clearly.
Done
Authors are suggested to include in materials and methods section how they have determined the dosage of yucca liquid for supplementation to animals. How do authors provide a statement that yucca liquid supplementation increases the nitrogen intestinal absorption in animals without proper experimental evidence. Authors should justify this.
The dose of yucca supplementation was given according to the previous studies and company recommendation. Again, we already stated the previous studies dealing with the same dose and we focused on the effect of yucca supplementation on gut microbes, antioxidant and immune status of birds. Regarding intestinal absorption, it is clear from the results of litter analysis that, nitrogen excretion was decreased gradually with increasing level of yucca supplementation which indicates the enhancement of intestinal absorption.
Does the excess nitrogen retention in poultry products effect human when consumed?
Excess nitrogen retention is important for both economy and clean environment. Regarding human health, I think it depends on many factors but it may be useful for human that production of poultry meat of high protein content may give us a new avenue to decrease the amount of meat consumption/individual/day which consequently affect positively on the national income.
I think, increasing nitrogen retention in poultry products will improve the public health and is a good strategy for humans.
Authors are suggested to include organoleptic tests to determine the quality of chicken or poultry products supplemented with yucca liquid for human consumption. Authors should discuss the role of antioxidant parameters and lipid peroxidation and compare their results with others work in discussion part.
All these parameters related to antioxidants have been discussed. Hope this form is suitable for publication.
Other citations are needed to strengthen the importance and relevance of the present results in the discussion section.
We added some relevant references in introduction and discussion to improve the paper quality…thanks a lot for your great efforts and help.
The format of references is not uniform. Need to check for typographical errors, plagiarism, punctuation, and grammar throughout the manuscript.
Done… several improvements have been made in the revised version….Hope this version is proper for publication.

Round 2
Reviewer 2 Report
I think the authors revised adequately their manuscript. I have no further comment.
Author Response
Thank you very much for your supportive comments and thanks a lot for your great efforts and help.
Reviewer 3 Report
In this manuscript, the authors have presented the results of their studies on the effect of Yucca liquid plus supplementation on poultry performance, biochemical parameters, gut microbial flora and nitrogen excretion. The revised version has been improved and the presented data are sound. However, there are still a few points that need to be addressed
Authors are suggested to include recent references in the manuscript instead of old references. Authors have included all the suggested references but at the wrong place. Suggested reference should be added at materials and methods section (Biochemical parameters especially for TBARS, GSH, CAT, and SOD)Following references are my suggestions to include instead of 21, 22, and 23 references.
Lavanya et al., 2012. Acetone Extract from Rhodomyrtus tomentosa: A Potent Natural Antioxidant. Evidence-Based Complementary and Alternative Medicine. 8, 2012.
Manubolu et al., 2014. Protective effect of Actiniopteris radiata (Sw.) Link. against CCl4 induced oxidative stress in albino rats. Journal of Ethnopharmacology. 153 (3): 744-752.
Lavanya et al., 2019. Protective Effects of Ammannia baccifera Against CCl4-Induced Oxidative Stress in Rats. International journal of environmental research and public health. 16 (8). 1440.
Does the deviated values (± values) in the table are standard errors or standard deviations (SE/SD)? Should be represented in table footnotes. Authors should discuss the role of antioxidant parameters and lipid peroxidation and compare their results with others work in discussion part. Other citations are needed to strengthen the importance and relevance of the present results in the discussion section. Once again check for the uniformity of references (according to journal format). Need to check for typographical errors, plagiarism, punctuation, and grammar throughout the manuscript once again.Author Response
In this manuscript, the authors have presented the results of their studies on the effect of Yucca liquid plus supplementation on poultry performance, biochemical parameters, gut microbial flora and nitrogen excretion. The revised version has been improved and the presented data are sound. However, there are still a few points that need to be addressed.
Thank you very much for your supportive comments and giving us a chance to improve our paper…thanks a lot for your great efforts and help.
Authors are suggested to include recent references in the manuscript instead of old references. Authors have included all the suggested references but at the wrong place. Suggested reference should be added at materials and methods section (Biochemical parameters especially for TBARS, GSH, CAT, and SOD)
Following references are my suggestions to include instead of 21, 22, and 23 references.
Lavanya et al., 2012. Acetone Extract from Rhodomyrtus tomentosa: A Potent Natural Antioxidant. Evidence-Based Complementary and Alternative Medicine. 8, 2012.
Manubolu et al., 2014. Protective effect of Actiniopteris radiata (Sw.) Link. against CCl4 induced oxidative stress in albino rats. Journal of Ethnopharmacology. 153 (3): 744-752.
Lavanya et al., 2019. Protective Effects of Ammannia baccifera Against CCl4-Induced Oxidative Stress in Rats. International journal of environmental research and public health. 16 (8). 1440.
Corrected as required
Does the deviated values (± values) in the table are standard errors or standard deviations (SE/SD)? Should be represented in table footnotes.
SE not SD. Added under the tables
Authors should discuss the role of antioxidant parameters and lipid peroxidation and compare their results with others work in discussion part.
Done as required
Other citations are needed to strengthen the importance and relevance of the present results in the discussion section.
We added some relevant references in introduction and discussion to improve the paper quality…thanks a lot for your great efforts and help.
Once again check for the uniformity of references (according to journal format). Need to check for typographical errors, plagiarism, punctuation, and grammar throughout the manuscript once again.
Done… several improvements have been made in the revised version….Hope this version is proper for publication.
